# Anti-Inflammatory Effects of Auranamide and Patriscabratine—Mechanisms and In Silico Studies

**DOI:** 10.3390/molecules27154992

**Published:** 2022-08-05

**Authors:** Kit-Kay Mak, Shiming Zhang, Jun Sheng Low, Madhu Katyayani Balijepalli, Raghavendra Sakirolla, Albena T. Dinkova-Kostova, Ola Epemolu, Zulkefeli Mohd, Mallikarjuna Rao Pichika

**Affiliations:** 1Department of Pharmaceutical Chemistry, School of Pharmacy, International Medical University, Kuala Lumpur 57000, Malaysia; kitkaymak@imu.edu.my (K.-K.M.); mohdzulkefeli@imu.edu.my (Z.M.); 2Centre of Excellence for Bioactive Molecules and Drug Delivery, Institute for Research, Development & Innovation, International Medical University, Kuala Lumpur 57000, Malaysia; 3School of Postgraduate Studies, International Medical University, Kuala Lumpur 57000, Malaysia; zhang.shiming@student.imu.edu.my (S.Z.); 00000022523@student.imu.edu.my (J.S.L.); 4Department of Pharmacology, Faculty of Medicine, MAHSA University, Jenjarom 42610, Malaysia; madhu@mahsa.edu.my; 5Department of Chemistry, Central University of Karnataka, Gulbarga 585367, India; srgoud2@gmail.com; 6Jacqui Wood Cancer Centre, Division of Cellular Medicine, School of Medicine, University of Dundee, Dundee DD1 4HN, UK; a.dinkovakostova@dundee.ac.uk; 7Departments of Medicine and Pharmacology and Molecular Sciences, School of Medicine, Johns Hopkins University, Baltimore, MD 21205, USA; 8Charles River Laboratories Edinburgh Ltd., Tranent EH33 2NE, UK; ola.epemolu@crl.com

**Keywords:** auranamide, patriscabratine, *Melastoma malabathricum*, NRF2, KEAP1, anti-inflammatory

## Abstract

Auranamide and patriscabratine are amides from *Melastoma malabathricum* (L.) Smith. Their anti-inflammatory activity and nuclear factor erythroid 2-related factor 2 (NRF2) activation ability were evaluated using *Escherichia coli* lipopolysaccharide (LPS*Ec*)-stimulated murine macrophages (RAW264.7) and murine hepatoma (Hepa-1c1c7) cells, respectively. The cytotoxicity of the compounds was assessed using a 3-[4,5-dimethylthiazol-2-yl]-2,5 diphenyl tetrazolium bromide (MTT) assay. The anti-inflammatory activity was determined by measuring the nitric oxide (NO) production and pro-inflammatory cytokines (Interleukin (IL)-1β, Interferon (IFN)-γ, tumour necrosis factor (TNF)-α, and IL-6) and mediators (NF-κB and COX-2). NRF2 activation was determined by measuring the nicotinamide adenine dinucleotide phosphate hydrogen (NADPH) quinone oxidoreductase 1 (NQO1), nuclear NRF2 and hemeoxygenase (HO)-1. In vitro metabolic stability was assessed using the mouse, rat, and human liver microsomes. The compounds were non-toxic to the cells at 10 μM. Both compounds showed dose-dependent effects in downregulating NO production and pro-inflammatory cytokines and mediators. The compounds also showed upregulation of NQO1 activity and nuclear NRF2 and HO-1 levels. The compounds were metabolically stable in mouse, rat and human liver microsomes. The possible molecular targets of NRF2 activation by these two compounds were predicted using molecular docking studies and it was found that the compounds might inhibit the Kelch domain of KEAP1 and GSK-3β activity. The physicochemical and drug-like properties of the test compounds were predicted using Schrodinger small molecule drug discovery suite (v.2022-2).

## 1. Introduction

Numerous reviews highlight the biological role of nuclear factor erythroid 2-related factor 2 (NRF2) in protecting human health and as a therapeutic target for various diseases [1,2]. Under basal conditions, NRF2 is negatively regulated by Kelch-like ECH-associated protein (KEAP1) via ubiquitination [3]. Under stress conditions, NRF2 detaches from KEAP1 in the cytoplasm [4]. Then, it translocates into the nucleus and forms a heterodimer with a small musculoaponeurotic fibrosarcoma (sMaf). The heterodimer activates the antioxidant response element (ARE) to produce a number of genes encoding for detoxifying enzymes, drug transporters, anti-apoptotic proteins and proteasomal subunits [5,6,7]. NAD(P)H: oxidoreductase 1 (NQO1) and hemeoxygenase (HO)-1 are the classical markers of NRF2 activation [8]. Because of the therapeutic importance of NRF2, many researchers in academia and industry have worked on developing drugs targeting NRF2 activation [1,8,9,10,11,12,13]. Only NRF2 activators (dimethyl fumarate (DMF) and its metabolite monomethyl fumarate (MMF)) are available for amyotrophic lateral sclerosis (ALS) [14]. The approval of these two drugs provides further impetus to discover improved drugs. DMF and MMF are electrophilic activators, thus lacking selectivity in their action. As outlined, many attempts have been made across the world to discover non-electrophilic NRF2 activators [1,12].

Natural products, especially plants, produce lead molecules in drug discovery. [15,16]. Many natural products, including sulforaphane, Bradoxolone methyl, curcumin, resveratrol, etc., are reported to activate NRF2 [17,18]. As part of our research on NRF2 activators, we identified *Melastoma malabathricum* (L.) Smith (Family: Melastomataceae), a well-known medicinal plant in Malaysia and Southeast Asia, which is a natural source for NRF2 activators. Our recent review article highlighted the biological effects (e.g., anti-inflammatory, wound healing, antiulcer, antimicrobial, etc.) and chemical constituents of *M. malabathricum* [19]. It contains many bioactive compounds, including flavonoids, tannins and amides [20]. Susanti et al. [21] reported quercetin, quercitrin, α-amyrin, patriscabratine, auranamide, and kaempferol-3-O-(2″,6″-di-O-p-trans-coumaroyl)glucoside from the leaves of *M. malabathricum* and their free radical scavenging, antioxidant and anti-inflammatory activities. Lestari et al. [22] determined the optimum extraction conditions of *M. malabathricum* leaves for the highest antioxidant activity with minimum toxicity. It is reported that hot water extract possesses the highest antioxidant activity, and it is more active than vitamin C, with minimum toxicity. In addition, they reported the presence of 4-O-caffeoylquinic acid, quercimeritin, digiprolactone, 3-O-trans-coumaroylquinic acid, norbergenin, and arteamisinin. Many studies reported the NRF2 activation of flavonoids [11]. The principal amides in *M. malabathricum* are auranamide and patriscabratine [23]. The in silico studies suggested the potential of these compounds to activate NRF2. The biological activities of these two compounds are not well explored except for a few reports on their anticancer activity. Thus, in this study, we investigated the NRF2 activation ability of the amides, their anti-inflammatory effects and mechanisms, and metabolic stability [24,25].

## 2. Results and Discussions

### 2.1. Cytotoxicity of Auranamide and Patriscabratine

The cytotoxicity of patriscabratine on cancer cells (gastric adenocarcinoma (AGS) and breast cancer (MDA-MB-231 and MCF-7)) was reported in the literature, and its IC_50_ value for these cells ranges from 69.8 to 197.3 μM [26]. However, there are no studies on the cytotoxicity of test compounds on normal cells. Thus, in this study, firstly, we determined the cytotoxicity of the test compounds on RAW264.7 and Hepa-1c1c7 cells using an MTT assay to determine the maximum non-toxic concentration that can be used to evaluate their biological activities in subsequent experiments. The percentage viability of the RAW264.7 and Hepa-1c1c7 cells is shown in Figure 1A,B. Both test compounds were non-toxic at a concentration of ≤10 μM to RAW264.7 and Hepa-1c1c7 cells. Thus, the test compounds’ anti-inflammatory activity and NRF2 activation were determined at 10, 1, 0.1, 0.01, and 0.001 µM.

### 2.2. Anti-Inflammatory Effect of Auranamide and Patriscabratine

The anti-inflammatory activity of the compounds was tested using *Escherichia coli* lipopolysaccharide (LPS*_Ec_*) stimulated RAW264.7 cells. Nitric oxide (NO) is a classical marker used to measure inflammation. The Griess assay was used to determine NO production [19]. The cells were stimulated with 100 ng/mL LPS*_Ec_* and the NO produced was considered to be at 100%. The anti-inflammatory effect of the compounds was assessed by adding the compounds (10, 1, 0.1, 0.01, and 0.001 µM) to the LPS*_Ec_* stimulated cells. The efficacy of the compounds was assessed by calculating their ability to reduce the LPS*_Ec_* stimulated NO production. The results (percentage NO production vs. concentration) are shown in Figure 1C. Dimethyl fumarate was used as a positive control, and 0.1% DMSO was used as a negative control. Both test compounds showed a concentration (10, 1 and 0.1 μM)-dependent effect in reducing the NO production. The anti-inflammatory effect was not observed at concentrations of less than 0.1 μM. The IC_50_ (concentration required to reduce the NO production by 50%) of auranamide is 1.22 ± 0.95 μM, and for patriscabratine it is 1.85 ± 0.93 μM. Auranamide (10 μM) and patriscabratine (10 μM) reduced the NO production by 33.22% and 30.84%, respectively.

### 2.3. Anti-Inflammatory Mechanisms of Auranamide and Patriscabratine

Pro-inflammatory cytokines (IL-6, IL-1β, IFN-γ and TNF-α) are produced innately as part of the host’s defence mechanism against inflammation. The excessive production of pro-inflammatory cytokines is the fundamental cause of many inflammatory diseases [24]. Most inflammatory processes are mediated via the activation of a key transcriptional regulator, the NF-κB subunit (p65) [27]. The pro-inflammatory cytokines induced Cyclooxygenase-2 (COX-2), a downstream enzyme in inflammation. Therefore, we tested the effect of the compounds on the expression of IL-6, IL-1β, IFN-γ, TNF-α, NF-κB and COX-2 using commercially available ELISA kits (Qiagen, Hilden, Germany). The activity of the test compounds was expressed as the fold change. The results (fold change vs. concentration) are shown in Figure 2. Dimethyl fumarate was used as a positive control, and 0.1% DMSO was used as a negative control. Both test compounds expressed a concentration (10, 1, and 0.1 μM)-dependent effect in reducing the levels of pro-inflammatory cytokines and the mediators indicated that LPS*_Ec_* upregulated the levels of pro-inflammatory cytokines IL-6 (6.91 ± 0.05), IL-1β (4.56 ± 0.13), IFN-γ (8.62 ± 0.26) and TNF-α (4.33 ± 0.17). Treatment with the compounds reversed the elevated levels of pro-inflammatory cytokines in a dose-dependent manner. The concentrations of the compounds from 0.001 to 10 μM resulted in the gradual inhibition of all pro-inflammatory cytokines where the compounds at 10 μM resulted in the highest reduction in fold change in IL-1β (auranamide, 2.05 ± 0.16; patriscabratine, 1.96 ± 0.12), IFN-γ (auranamide, 4.06 ± 0.19; patriscabratine, 4.15 ± 0.21), TNF-α (auranamide, 2.42 ± 0.24; patriscabratine, 2.53 ± 0.18), and IL-6 (auranamide 2.85 ± 0.31; patriscabratine, 2.51 ± 0.38). Overall, the results showed that treatment with the compounds reversed the upregulated levels of pro-inflammatory cytokines IL-6, IL-1β, IFN-γ and TNF-α.

The treatment with the compounds dose-dependently inhibited the expression of COX-2 and NF-κB. It is apparent that the treatment of cells with the compounds at 10 μM, resulted in 2.53 ± 0.25 (auranamide) and 2.17 ± 0.31 (patriscabratine) fold changes (LPS*_Ec_* control: 8.09 ± 0.08) in COX-2; and 2.13 ± 0.20 (auranamide) and 2.25 ± 0.27 (patriscabratine) fold changes (LPS*_Ec_* control: 8.40 ± 0.09) in NF-κB p65.

### 2.4. The Effect of Auranamide and Patriscabratine on NRF2 Activation

The NRF2 activation is associated with increased NQO1 activity and an increased concentration of NRF2 and HO-1 in the nucleus. The protective role of NRF2/HO-1 in inflammatory diseases is well documented [28,29]. Therefore, the NRF2 and HO-1 levels in nuclear extracts of the RAW 264.7 cells were determined using ELISA kits and expressed as fold changes with reference to the negative control. The effect of test compounds on NQO1 activity was determined using the reported method [30,31] in murine hepatoma cells (Hepa-1c1c7). Dimethyl fumarate was used as a positive control, and 0.1% DMSO was used as a negative control. The results are summarised in Figure 3. Both test compounds showed a concentration-dependent effect in elevating the NRF2 and HO-1 levels and NQO1 activity. The CD value (concentration of test compound to double the NQO1 activity) of auranamide was 6.81 ± 0.18 μM, while for patriscabratine it was 4.28 ± 0.17 μM.

As shown in Figure 3, auranamide and patriscabratine treatment resulted in NRF2 and HO-1 upregulation in a concentration-dependent manner. The treatments at 10 μM upregulated NRF2 by 2.66 ± 0.16 (auranamide) and 2.37 ± 0.06 (patriscabratine)-fold; and HO-1 by 1.74 ± 0.09 (auranamide) and 1.83 ± 0.07 (patriscabratine)-fold.

### 2.5. Metabolic Stability of Auranamide and Patriscabratine in Liver Microsomes

Metabolic stability is an important parameter in the drug discovery pipeline [32]. The metabolic stability of test compounds was evaluated in vitro using human, rat and mouse liver microsomes. The metabolic equation below determines the test compounds’ half-life (T_1/2_) and intrinsic clearance (Cl_int_).

$$\mathrm{Compound}+\;\mathrm{NADPH}\;+{\;H}^{+}+{\;O}_{2}\;\to \;\mathrm{oxidized}\;\mathrm{analyte}\;+{\;\mathrm{NADP}}^{+}+{\;H}_{2}O$$


The results are tabulated in Table 1. Both test compounds showed rapid intrinsic clearance in all three microsomal enzymes (>5 mL/min/g liver) [32,33]. The results indicate that the compounds are quickly metabolized despite their large molecular structure (auranamide MW: 506.6 g/mol; patriscabratine MW: 444.5 g/mol).

Based on the results, both compounds’ half-life (T_1/2_) was found to be relatively consistent in the human microsomes, whereas auranamide falls in the 4 min range, and patriscabratine falls in the late 3 min range in rodents (rats and mice).

### 2.6. In Silico Studies

The in silico studies were carried out using the Schrodinger small molecule drug discovery suite (Version 2022-2). Molecular docking studies were conducted to determine the possible mechanisms of test compounds in NRF2 activation. There are three main mechanisms through which NRF2 activation occurs, namely (a) covalent bonding with Cys residues (electrophilic activators), (b) inhibition of Kelch domain of KEAP1, and (c) inhibition of GSK-3β activity. Covalent binding studies indicate that the test compounds are not electrophilic activators. Molecular docking and MM-GBSA studies suggested that the test compounds form a stable interaction with the Kelch domain of KEAP1 (PDB ID: 4IQK) and GSK-3β (PDB ID: 3ZRL). The interactions (2D and 3D) between the test compounds and the receptors’ binding sites are shown in Table 2. With the Kelch domain of KEAP1, auranamide and patriscabratine form H-bonding interactions with Ser 555 and Ser 508. With the GSK-3β enzyme, auranamide forms an H-bonding interaction with Lys 183, while patriscabratine forms only hydrophobic interactions.

The docking scores (XP docking) and binding energies (MM-GBSA calculations) of the test compounds with the Kelch domain of KEAP1 and GSK-3β are shown in Table 3. All the values in the table are presented in Kcal/mole. The negative docking scores and negative binding energies (ΔG) suggested stable interactions between the test compounds and protein binding sites; thus, their NRF2 activation was mediated via the modulation of the activity of both proteins. Generally, the docking scores and binding energies (ΔG bind) with 4IQK were more negative than those of 3ZRL, indicating that the test compounds favour binding to the Kelch domain over GSK-3β. Thus, it could be deduced that NRF2 activation by these two compounds is mediated predominantly through the Kelch domain. The binding energy values (Table 3) show that the stable interactions between the test compounds and binding sites are predominantly mediated via Van der Waals interactions followed by coulombic and lipophilic interactions.

As shown in Table 4, the physicochemical and drug-like properties of the test compounds were predicted using QikProp wizard.

The compound auranamide violated two of the rules of (mass and logPo/w) of Lipinski’s rule of 5, whereas patriscabratine obeyed all the rules. The ideal compounds were not expected to have CNS activity in a drug discovery cycle. Both test compounds were predicted to be CNS inactive. Compounds with ideal permeability properties are highly sorted in the drug discovery pipeline, and both test compounds were predicted to have ideal cell permeability properties. Additionally, ideal compounds were expected to have a low binding affinity to serum albumin. Patriscabratine has a very low affinity to serum albumin compared to auranamide. Next, estimating the effect of the compound on cardiotoxicity is also an essential criterion in the early phase of drug discovery. Both test compounds were predicted (logHERG) to be non-cardiotoxic. Other critical factors in determining the drug-like properties of a compound are the polar surface area and oral absorption. Both test compounds were predicted to have a desirable polar surface area and good oral absorption. These predictions suggest that both compounds have a high potential to be drug-like; however, patriscabratine is predicted to be better than auranamide.

## 3. Materials and Methods

Murine macrophages (RAW 264.7, #ATC.TIB-71) and murine hepatoma cells (Hepa-1c1c7, #ATC.CRL-2026) were purchased from American Type Culture Collection (ATCC, USA). Unless otherwise specified, all cell culture reagents were purchased from Sigma Aldrich, St. Louis, MO63118, USA. ATCC protocols were followed to culture the cells, and the cells from passages 5 to 20 were used in all the experiments. ACCUTASE^TM^ cell detachment solution was used to detach the cells. Auranamide and patriscabratine (95% purity by LC-MS) were extracted from the methanol extract of *M. malabathricum* using flash pressure liquid chromatography (Reveleris^®^ amino12g flash Cartridge, Reveleris^®^X2, BÜCHI Labortechnik AG, 9230 Flawil, Switzerland). Their chemistry was confirmed using NMR and LC-MS (Details of the extraction, isolation, NMR spectrum and LC-MS spectrum are provided in Appendix A). The MTT Cell Proliferation Assay Kit (Vybrant^®^, #V13154) was purchased from Thermo Fisher Scientific, Waltham, MA, USA. Griess reagent system (#G2930) was purchased from Promega Corporation, USA. The cytokine ELISA kits (#SEM03109A, IL-1β; #SEM03015A, IL-6; #SEM06411A, TNF-α; and #SEM03121A, IFN-γ) were purchased from Qiagen, Hilden, Germany. The nuclear extraction kit (#113474), Bradford’s reagent (#102535), COX-2 ELISA kit (#210574) and HO-1 ELISA kit (#204524) were purchased from Abcam, Cambridge, UK. NF-κB p65 transcription factor assay ELISA kit (#E-EL-M0838) was purchased from Elabscience, Houston, TX, USA. NRF2 ELISA kit (#OKCD09342) was purchased from Aviva Systems Biology, San Diego, CA 92121, USA. Pooled human liver microsomes, mixed-gender (HMMCPL), pooled male CD-1 mouse liver microsomes (MSMCPL), and pooled Sprague Dawley rat liver microsomes (RTMCPL) were purchased from ThermoFisher Scientific, Waltham, MA, USA.

### 3.1. Preparation of Auranamide and Patriscabratine Solutions

The required amount of auranamide and patriscabratine were dissolved in molecular biology grade DMSO to prepare a 10 mM solution, which was further diluted with phosphate-buffered saline (PBS, pH 7.4) to obtain the required concentrations. In all in vitro experiments, the final concentration of DMSO was not more than 0.1% (*v*/*v*).

### 3.2. Cytotoxicity of Auranamide and Patriscabratine

The cytotoxicity of the test compounds on RAW 264.7 and Hepa-1c1c7 cells was assessed using Vybrant^®^ MTT Cell Proliferation Assay Kit following the manufacturer’s protocol. The cells were seeded at 5 × 10^5^ cells/well and incubated for 24 h. Then, various concentrations (100 to 0.01 μM) of the test compounds were added and incubated for a further 24 h. The cell supernatant solution was replaced with the MTT assay solution and incubated for 4 h. The formazan crystals were dissolved in DMSO, and the optical density (OD) of the solution was measured at 570 nm (reference wavelength 630 nm) using Spectramax M3 microplate reader (Molecular Devices, LLC., San Jose, CA 95134, USA). The cell viability was calculated using the following formula, and the results are presented as mean ± SD.

$$\mathrm{Percent}\;\mathrm{cell}\;\mathrm{viability}\;\left(\%\right)=\left(\frac{{\mathrm{OD}}_{\mathrm{treated}\;\mathrm{cells}}-{\mathrm{OD}}_{\mathrm{blank}}}{{\mathrm{OD}}_{\mathrm{untreated}\;\mathrm{cells}}-{\;\mathrm{OD}}_{\mathrm{blank}}}\right)\times 100\%$$


In all further experiments, auranamide and patriscabratine in the concentration range of 10–0.001 µM (1:10 dilution) were tested as they were found to be cytotoxic at concentrations higher than 10 µM.

### 3.3. In Vitro Anti-Inflammatory Activity of Auranamide and Patriscabratine

The anti-inflammatory activity of the test compounds was assessed in *E. coli* lipopolysaccharide (LPS_Ec_)-induced inflammation in RAW 264.7 cells. Cells (5 × 10^5^ cells/well) were incubated for 24 h in a 24-well plate. Then, the cells were challenged with LPS_Ec_ (100 ng/mL) for 4 h. The test compounds (10, 1, 0.1, 0.01 and 0.001 µM) or positive control (dimethyl fumarate, 10 μM) were then added and incubated for 24 h. The levels of NO production and pro-inflammatory cytokines (IL-1β, IL-6, IFN-γ and TNF-α) were determined using the supernatant solution. The cell pellet was used to measure the levels of COX-2, NF-κB p65, NRF2, and HO-1.

### 3.4. Effect of Auranamide and Patriscabratine on NO Production in LPSEc Challenged RAW 264.7 Cells

LPS*_Ec_* activates inducible nitric oxide synthase (iNOS) in RAW 264.7 cells and produces nitric oxide (NO), an important mediator of inflammation [18]. The NO content in the supernatant solution was measured using the Griess assay following the manufacturer’s instructions. The optical density (OD) was measured at 540 nm using a Spectramax M3 microplate reader (Molecular Devices, LLC., San Jose, CA 95134, USA). The formula provided below was used to calculate the percentage of NO production.

$$\mathrm{Percent}\;\mathrm{NO}\;\mathrm{production}=\left(\frac{{\mathrm{OD}}_{\mathrm{treated}\;\mathrm{cells}}-{\;\mathrm{OD}}_{\mathrm{blank}}}{{\mathrm{OD}}_{\mathrm{untreated}\;\mathrm{cells}}-{\;\mathrm{OD}}_{\mathrm{blank}}}\right)\times 100\%$$


The activity of the test compounds was tested in triplicate on each plate. The results are presented as mean ± SD. Treatment was performed with either LPS*_Ec_* alone or LPS*_Ec_* and test compounds were designated as treated cells. RAW264.7 cells treated with vehicle (0.1% DMSO) were designated as untreated cells.

### 3.5. Effect of Auranamide and Patriscabratine on Pro-Inflammatory Cytokines Expression in LPS_Ec_ Challenged RAW 264.7 Cells

Single-Analyte ELISArray kits were used to quantify the pro-inflammatory cytokines (IL-1β, IL-6, IFN-γ and TNF-α) in the cell supernatant following the manufacturer’s instructions. The standard protein (50 μL) or test sample solutions (50 μL) were added to a well (in duplicate) and incubated for 2 h at room temperature. The contents in the well were washed with wash buffer, and then 100 µL of biotinylated detection antibody solution was added and incubated for 1 h at room temperature. The contents in the wells were washed again with wash buffer. Then, an avidin-horseradish peroxidase conjugate solution (100 µL) was added and incubated in the dark for 30 min. Again, the wash buffer was used to wash away the unbound material. One hundred microliters of development solution was added to the wells and incubated in the dark for 15 min, followed by the addition of stop solution (100 µL) and incubation was performed in the dark for 10 min. The optical density was measured at 450/570 nm using a Spectramax microplate reader M3 (Molecular Devices, LLC., San Jose, CA 95134, USA). The levels of pro-inflammatory cytokines (in pg/mL) in test samples were calculated using the standard curve. The below formula was used to calculate the fold change, and the results are presented as mean ± SD (n = 3).

$$\mathrm{Fold}\;\mathrm{change}=\frac{\;\mathrm{cytokine}\;\mathrm{level}\;\mathrm{in}\;\mathrm{LPS}Ec\;\mathrm{or}\;\mathrm{compound}\;\mathrm{treated}\;\mathrm{cells}\;}{\mathrm{cytokine}\;\mathrm{level}\;\mathrm{in}\;\mathrm{untreated}\;\mathrm{cells}}$$


### 3.6. Effect of Auranamide and Patriscabratine on COX-2, NF-κB p65, NRF-2 and HO-1 Expression in LPS_Ec_-Stimulated RAW 264.7 Cells

The COX-2 level was estimated in the cytoplasmic extract, whereas NF-κB p65, NRF-2 and HO-1 levels were estimated in nuclear extract. The nuclear extraction kit was used to separate the cytoplasm from the nucleus following the manufacturer’s instructions. The cell pellet was incubated in 100 µL of pre-extraction buffer on ice for 10 min, vortexed, and centrifuged. The supernatant cytoplasmic extract was collected. The nuclear pellet was incubated in an extraction buffer containing Dithiothreitol (DTT) and Protease Inhibitor Cocktail (PIC) on ice for 15 min. The suspension was centrifuged to obtain the nuclear extract. The protein concentration in the extracts was determined using Bradford’s reagent. The respective ELISA kits were used to measure the levels of COX-2, NF-κB p65, NRF-2 and HO-1. The levels were measured in triplicate. The absorbance was measured using a Spectramax M3 microplate reader (Molecular Devices, LLC., San Jose, CA 95134, USA). The effect of the test compounds on the expression was represented as fold change with reference to the negative control, using the following formula, and the results are presented as mean ± SD (n = 3).

$$\mathrm{Fold}\;\mathrm{change}=\;\frac{\mathrm{Protein}\;\mathrm{level}\;\mathrm{in}\;\mathrm{LPS}Ec\mathrm{or}\;\mathrm{compound}\;\mathrm{treated}\;\mathrm{cells}\;}{\mathrm{Protein}\;\mathrm{level}\;\mathrm{in}\;\mathrm{untreated}\;\mathrm{cells}}$$


### 3.7. Effect of Auranamide and Patriscabratine on NQO1 Activity in Hepa-1c1c7 Cells

The “Prochaska” Microtiter Plate Bioassay was used to measure the NQO1 activity [34] in hepatoma cells (Hepa-1c1c7). The cells (5 × 10^5^ per well) were seeded and incubated with either auranamide or patriscabratine (10, 1, 0.1, 0.01, and 0.001 µM) or a reference drug (dimethyl fumarate, 10 μM) for 48 h. The cells were washed with PBS and then lysed with digitonin solution (75 µL). The protein concentration was determined using Bradford’s reagent. The cell lysate was incubated with a reaction mixture (0.5 M Tris-Cl buffer, 7.5 mM FAD, 150 mM glucose-6-phosphate, 2 U/mL glucose-6-phosphate dehydrogenase, 50 mM NADP+, 25 mM menadione and 0.7 mM MTT) at 37 °C for 5 min. The optical density of the solution was determined at 610 nm using a Spectramax M3 microplate reader (Molecular Devices, LLC., San Jose, CA 95134, USA), and the values were normalized to the total protein content.

### 3.8. Metabolic Stability of Auranamide and Patriscabratine in Liver Microsomes

The metabolic stability of the test compounds in human, rat, and mouse liver microsomes was determined using the method described in our previous work [8]. Five microlitres of auranamide or patriscabratine (0.5 µM) were incubated in a reaction mixture at 37 °C for 0, 3, 6, 15, 30, 45 and 60 min. The concentration of test compounds in the reaction mixture was quantified using Agilent 1290 coupled with Q-TOF. The experiment was carried out in triplicate. The concentration versus time plot calculated the elimination rate constant (K_el_). The below formulae were used to calculate the intrinsic clearance (Cl_int_) and half-life (T_1/2_) [35]. The results are expressed as mean ± standard deviation from three experiments.

$${\mathrm{Cl}}_{\mathrm{int}}=\;\frac{K_{\mathrm{el}}}{0.5}\;\times 52.5$$


$$T_{1/2}=\;\frac{0.693}{K_{\mathrm{el}}}$$



### 3.9. In Silico Studies

In silico studies with test compounds were conducted using Schrödinger small molecule drug discovery suite (v.2022-2, Schrödinger, LLC, New York, NY, USA). The default setting was applied unless otherwise specified. The chemical structures of the test compounds were sketched (using a 2D sketcher) and imported into Maestro. The test compounds were minimized using LigPrep wizard (Force field, OPLS4; pH, 7.2 ± 0.2). Two important mechanisms of NRF-2 activation by non-electrophilic activators are the inhibition of the Kelch domain of KEAP1 and inhibition of GSK-3β enzyme. The crystal structures of the Kelch domain (PDB ID: 4IQK) and GSK-3β (PDB ID: 3ZRV) were downloaded from the RCSB PDB website. The crystal structures were prepared using the protein preparation wizard (Force field, OPLS4; pH, 7.2 ± 0.2). Monomers of these crystal structures were used for molecular docking studies carried out with the GLIDE XP protocol. The test compounds’ glide G (docking score) was recorded. The 2D-interaction and 3D-interaction diagrams were captured. The binding energy of the test compounds with the proteins was calculated using the MM-GBSA protocol. The physicochemical and drug-like properties of the test compounds were determined using the QikProp module.

## 4. Conclusions

This is the first study to report the anti-inflammatory activity of auranamide and patriscabratine and their mechanisms. Both compounds significantly downregulated the NO production, pro-inflammatory cytokines (IL-1β, IL-6, IFN-γ and TNF-α), and inflammatory mediators (COX-2 and NF-κB p65) in LPS*_Ec_* stimulated RAW 264.7 cells. The compounds also upregulated the NRF2 and HO-1 protein in the nuclear extracts of RAW 264.7 cells. In addition, the test compounds upregulated the NQO1 enzyme activity. Collectively, these observations suggest that both compounds’ anti-inflammatory activity is mediated via NRF2 activation. There are numerous studies which report the association of NRF2 activation with beneficial effects in inflammatory, neurological, and metabolic diseases. Thus, these two compounds may have significant therapeutic effects on the diseases mentioned above. Additionally, the multiple therapeutic uses of *M. malabathricum* in folk medicine for managing various disease conditions could be due to its ability to activate NRF2. In a drug discovery pipeline, metabolic stability is important in drug metabolism and pharmacokinetics studies. Despite their heavy molecular weight, these two compounds were rapidly cleared in human, rat, and mouse liver microsomes. In addition, these two compounds’ physicochemical and drug-like properties were found to be in the favourable range of oral drug-like molecules.

NRF2 activation can occur either via KEAP1-dependent or independent pathways. The KEAP1-dependent pathway involves either covalent bond formation with cysteine residues (electrophilic activators) or the inhibition of the Kelch domain (non-electrophilic activators). In general, non-electrophilic activators are preferred over electrophilic activators mainly because the latter are non-specific in their action. The molecular docking and MM-GBSA studies revealed that these two compounds form stable interactions with the Kelch domain, thereby suggesting these two compounds are non-electrophilic NRF2 activators. KEAP1-independent NRF2 activation involves the inhibition of GSK-3β enzyme activity. Molecular docking and MM-GBSA studies revealed that these two compounds form stable interactions with GSK-3β, suggesting that these two compounds may also activate NRF2 via KEAP-1 independent pathways. In summary, auranamide and patriscabratine, in the sub-micromolar range, possess significant anti-inflammatory activity, likely, via KEAP1-dependent (non-electrophilic) and KEAP1-independent NRF2 activation. Since these two compounds possess oral drug-like properties and are metabolically stable, their pharmacological activities and molecular mechanisms should be further explored physiologically and in disease-relevant test models.

## Figures and Tables

**Figure 1 molecules-27-04992-f001:**
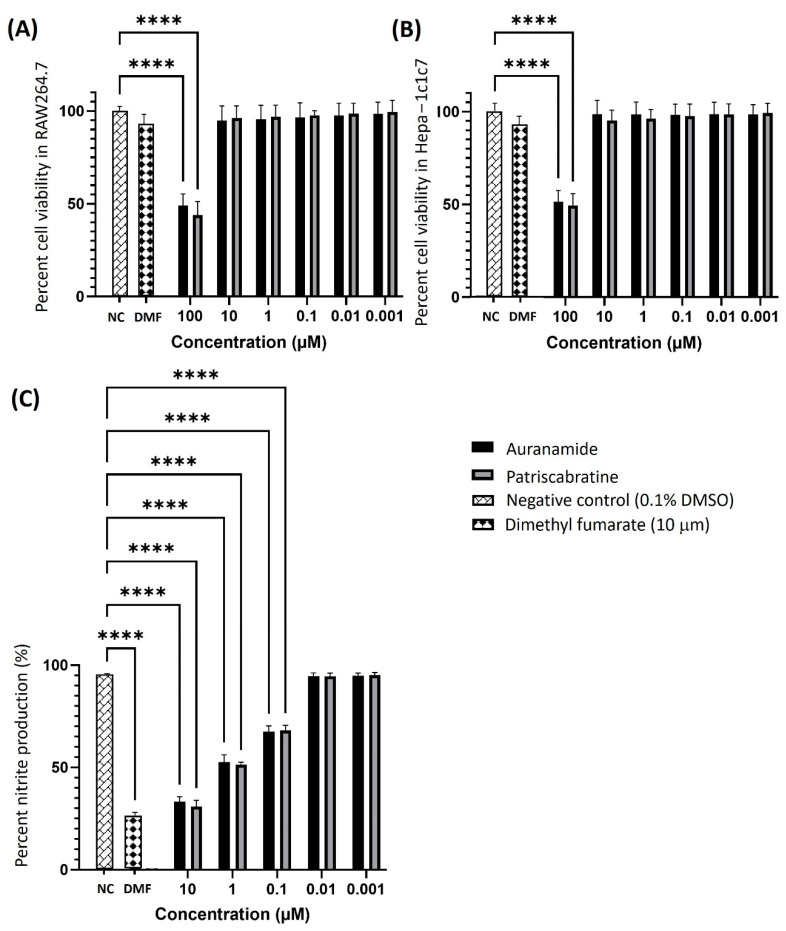
The cytotoxicity of test compounds on (**A**) murine macrophages (RAW 264.7); (**B**) murine hepatoma (Hepa-1c1c7) cells; (**C**) the anti-inflammatory activity of test compounds in LPS*Ec* challenged RAW 264.7 cells. The results are expressed as the mean ± SD (n = 3). ****, *p* ≤ 0.0001 compared to LPS*_Ec_* treated cells. The bars without annotation indicate the values are not significant with reference to the negative control.

**Figure 2 molecules-27-04992-f002:**
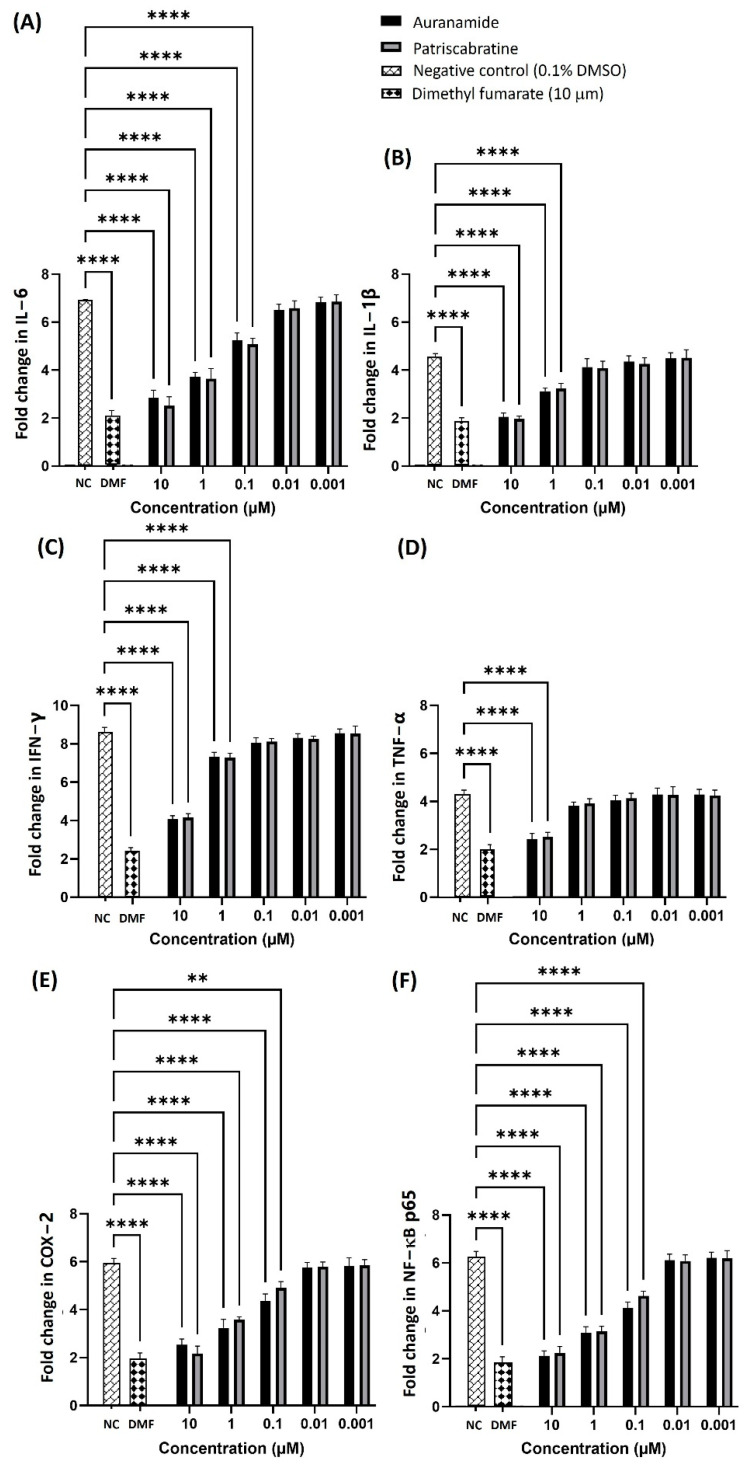
The effect of auranamide and patriscabratine on pro-inflammatory cytokines (**A**) IL-6, (**B**) IL-1β, (**C**) IFN-γ, and (**D**) TNF-α; and mediators (**E**) COX-2 and (**F**) NF-κB. The activity of the test compounds was expressed as fold change. The results are expressed as mean ± SD (n = 3). **, *p* ≤ 0.01; ****, *p* ≤ 0.0001 compared to LPS*_Ec_* treated cells. The bars without annotation indicate the values are not significant with reference to the negative control.

**Figure 3 molecules-27-04992-f003:**
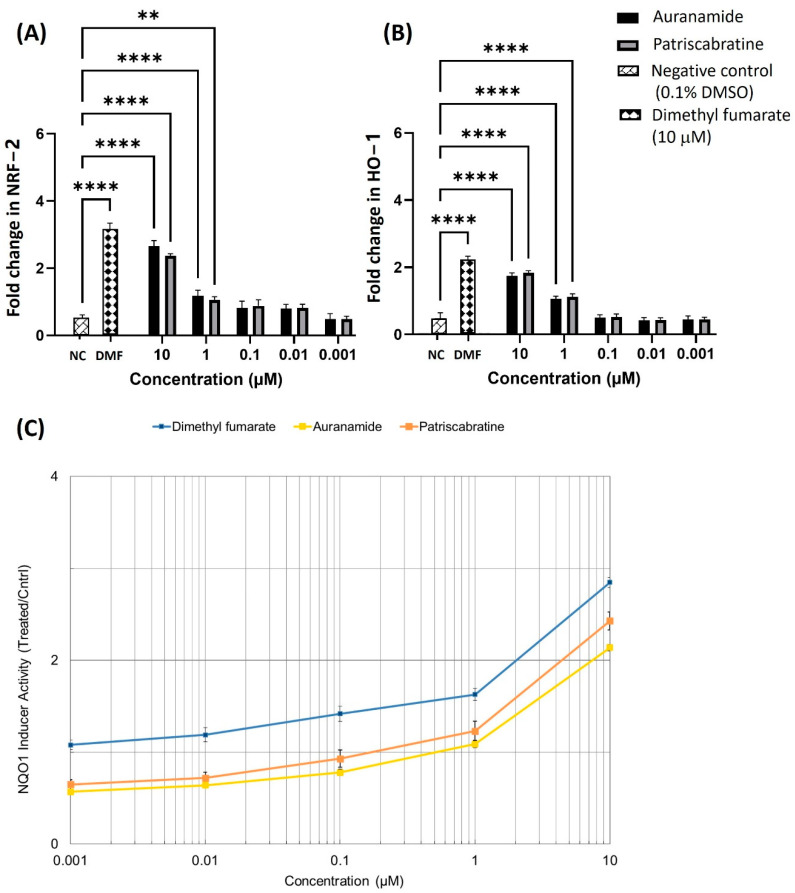
The effect of auranamide and patriscabratine on (**A**) NRF2 protein expression; (**B**) HO-1 protein expression; (**C**) NQO1 activity in whole cell lysate of Hepa-1c1c7 cells. The results are expressed as mean ± SD (n = 3). **, *p* ≤ 0.01; ****, *p* ≤ 0.0001 compared to LPS_Ec_ + 0.1% DMSO treatment.

**Table 1 molecules-27-04992-t001:** Metabolic stability profile of auranamide and patriscabratine in human, rat and mouse liver microsomes.

	Compound	Human	Rat	Mouse
Cl_int_ (mL/min/g liver)	Auranamide	10.18 ± 0.35	17.22 ± 0.46	16.18 ± 0.43
	Patriscabratine	13.27 ± 0.42	18.53 ± 0.56	18.64 ± 0.51
Half-life (T_1/2_, min)	Auranamide	7.14	4.23	4.50
	Patriscabratine	5.48	3.93	3.90

Note: The metabolic stability results in human, rat and mouse liver microsomes are presented as mean ± SD (n = 3). The compounds are rapidly cleared as their in vitro experimental values fall in the high clearance classification band for humans, rats and mice (high clearance: >5 mL/min/g liver; low clearance: <5 mL/min/g liver) [32,33].

**Table 2 molecules-27-04992-t002:** The two-dimensional (2D) and three-dimensional (3D) images of docking studies of auranamide and patriscabratine in KEAP1 Kelch domain (PDB ID: 4IQK) and GSK-3beta protein (PDB ID: 3ZRL).

PDB	CMP	2D	3D
4IQK	AUR	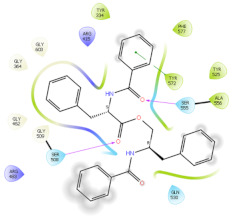	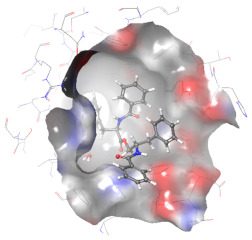
4IQK	PAT	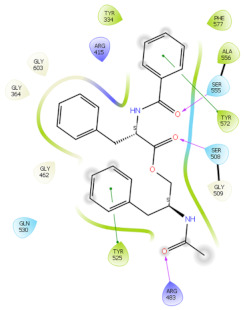	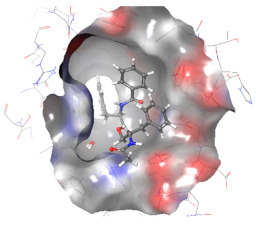
3ZRL	AUR	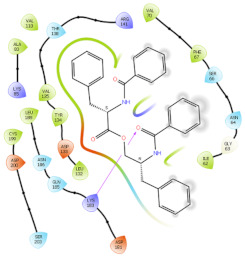	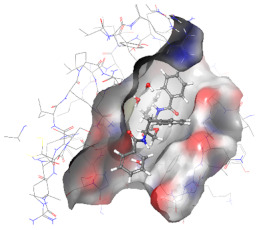
3ZRL	PAT	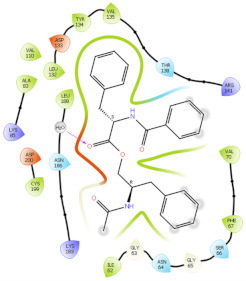	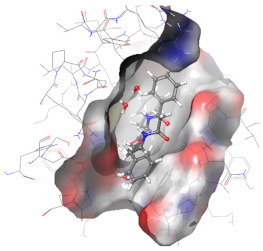

Note: AUR, Auranamide; PAT, Patriscabratine.

**Table 3 molecules-27-04992-t003:** The docking scores and binding energies of auranamide and patriscabratine in KEAP1 Kelch domain (PDB ID: 4IQK) and GSK-3beta (PDB ID: 3ZRL). The values are in kcal/mole.

PDB	Cpd	Docking Score	ΔG Bind	ΔG Coulomb	ΔG Covalent	ΔG Hbond	ΔG Lipo	ΔG VdW
4IQK	AUR	−6.768	−72.34	−21.68	2.09	−1.05	−24.17	−50.48
	PAT	−6.167	−69.44	−23.39	2.10	−1.19	−21.96	−48.61
3ZRL	AUR	−5.518	−49.69	−17.57	6.23	−1.62	−22.12	−52.74
	PAT	−5.270	−45.50	−21.71	13.30	−1.09	−25.05	−51.89

Note: Cpd, compound; AUR, auranamide; PAT, patriscabratine; Hbond, hydrogen bond; Lipo, lipophilic; VdW, Van der Waals.

**Table 4 molecules-27-04992-t004:** The representative physicochemical properties and drug-like properties of auranamide and patriscabratine.

Cpd	CNS	Mass	logPo/w	logS	logHERG	PCaco	logKhsa	%HOA	PSA	RO5
AUR	−2	506.6	7.149	−8.1	−8.622	1335	1.373	100	101	2
PAT	−2	444.5	4.836	−5.2	−5.807	425	0.545	100	102	0

Note: Cpd, compound; AUR, auranamide; PAT, patriscabratine; CNS, predicted central nervous system activity (−2 inactive, 2 active); logPo/w, predicted octanol/water partition coefficient (−2.0 to 6.5); logS, predicted water solubility (−6.5 to 0.5); logHERG, predicted IC_50_ value for blockage HERG K^+^ channels (concern below −5); Pcaco, predicted Caco-2 cell permeability (<25 poor, >500 great); logKhsa, prediction of binding to human serum albumin (−1.5 to 1.5); %HOA, percent human oral absorption (>80% is high, <25% is poor); PSA, polar surface area (7.0 to 200); RO5, number of violations of Lipinski’s rule of five (maximum is 4).

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
