# Peer review of "Anti-Inflammatory Effects of Auranamide and Patriscabratine—Mechanisms and In Silico Studies"

_molecules, 2022, doi:10.3390/molecules27154992_

Round 1
Reviewer 1 Report
General comments
This paper was well written and, to some extent, has some innovations. Based on a series of experiments, authors found for the first time that Uranamide and Patriscabratine have significant anti-inflammatory effects, and most likely via KEAP-1 dependent (non-electrophilic) and KEAP-independent NRF2 activation. It has important value in scientific research. However, the research is not rich of enough experiment documents to support explain of anti-inflammation mechanism of the drugs, metabolic stability of drug is not closely correlative with anti-inflammation mechanism, and molecular docking research just provide a theoretical possibility without experiment documents. Moreover, there are also many mistakes or errors throughout of the paper.
Specific comments
1. Some abbreviations, such as Nrf2, LPSes, NQO1, Ho-1, KEAP1, should appear with their full names when first appearing in the abstract rather than in straight matter.
2. The abstract is necessary to add the related content of the in Silico studies, which is consistent with the title and research content.
3. Line 46 “.....encoding for.” should delete the full stop, and the last of line 47 also has a excrescent punctuation that is at the frond of the reference 8.
4. Line 75-77, the viability of the cells were described as 94.87 ± 7.82 μM, I don’t think its correction. It should be rechecked.
5. Line 166, the word “shows” ma be unnecessary.
6. Line 177-178, “IL-6 (6.91 ± 0.05), IL-1β (4.56 ± 0.13), IFN- γ (8.62 ± 0.26) and TNF-α (4.33 ± 0.17)” are values of control negative of each index, not the “LPSEC upregulated”. Meanwhile, I understand that each value with SD represents the fold change of the index level, but it doesn’t find any normalized group in figure 2. So, which group is set as normalized group for the fold change calculation? The problem is also involved in figure 3.
7. Line 177-181, what’s the standard of the clearance classification ?
8. Regarding to interaction between both drugs and KEAP1 and GSK-3β, why don’t authors conducted experiments to prove it such as western blot or surface plasmon resonance, while authors made molecular docking studies. I hold that research experiments are more reliable than a prediction. Moreover, there is no any discussion on drug mechanism based on correlation between the drugs and the function changes of NRF2, KEAP, and GSK-3β.
9. When inserting a picture, "Figure and the number immediately following" should not be the subject of the comment, and "Figure and the figures immediately following" should be bold and separated from the following text by ". "; When inserting a Table, "Table and the number immediately following" should be bold and separated from the following text by ". ".
10. In Figure 1, units should be added to the Y-axis of figures A) and B).
11. In the comment section of Figure 2, it is suggested to add the sentence "The activity of the test compounds was expressed as the fold change" for subsequent readers.
12. In the annotation section of Figure 3, it is suggested to add annotations for each of the three images to facilitate subsequent readers.
13. In Table 1, time units should be added to the “Half-life” column.
14. In Table 4, no ordinal number and comment are added, and corresponding ordinal number and comment need to be added.
15. Line 404 "keAP1-dependent" should be changed to "keap1-dependent" to be consistent with the other lines.
16. The Conclusion is too longer and superfluous, which is more like discussion.
Author Response
Thanks for the comments. Our response to the comments is attahced.

Reviewer 2 Report
The manuscript “Anti-inflammatory Effects of Auranamide and Patriscabratine – Mechanisms and in Silico Studies” can be of interest to wide readers of journals and contributes to existing knowledge on the subject matter. However, I have pointed out few pertinent points for improving the clarity of the content and boosting the scientific soundness of the manuscript.Abstract:
Write the plant name as given below.
Melastoma malabathricum (L.) Smith
Line 23: Include the family name of the plant.
Line 26: Write the full form of MTT and NO.
Introduction:
Write more about the medicinal values of the Melastoma malabathricum (L.) Smith
Line 48: Define abbreviation when used first time
“NAD(P)H “
Line 50: “many researchers”… Give multiple references to support the statements.
Results:
Line 82-87: More suitable in the Material and methods section
Line 100: Write just (C). Not “Figure C”
Discussion of the results obtained is completely missing in the following two sub-headings. Use sufficient citations to discuss and support the results mentioned in the following sub-headings
2.1. Cytotoxicity of auranamide and patriscabratine
2.2. Anti-inflammatory effect of auranamide and patriscabratine
Poor discussion. Use sufficient citations to discuss and support the results mentioned in the following sub-heading
2.3. Anti-inflammatory mechanisms of auranamide and patriscabratine
Figure 3: Cntrl stands for? Write full form.
Line 140: Check the format
(23) (24).
Poor discussion. Use sufficient citations to discuss and support the results mentioned in the following sub-headings
2.4. The effect of auranamide and patriscabratine on NRF2 activation
2.5. Metabolic stability of auranamide and patriscabratine in liver microsomes
Table 2: both 2D and 3D pictures are not clear. Replace it with a more distinct one.
Poor discussion. Authors need to strengthen the discussion section by adding more interpretations of recorded findings supported by peer-findings.
2.6. In silico studies
Line 251: Write detail about the methods used to obtain the “Auranamide and patriscabratine” in the experiment.
Conclusion
Need to present concrete findings based on recorded data, and avoid using too generalized statement.
Author Response

(The authors gave the same response as above.)

Round 2
Reviewer 1 Report
I have no suggestion to authors.
Author Response
There are no suggestions made.
Reviewer 2 Report
There are some minor issues in the manuscript. The author of the manuscript need to revise the text
Line 258: Sample extraction procedure and conditions in detail.
Please provide MRM, chromatogram of LC/MS for this experiment. It will be useful for other researchers.
Discussion of the results obtained is completely missing in the following two sub[1]headings. Use sufficient citations to discuss and support the results mentioned in the following sub-headings. There are many recent works/literatures related to auranamide and patriscabratine.
2.1. Cytotoxicity of auranamide and patriscabratine.
2.2. Anti-inflammatory effect of auranamide and patriscabratine
Read the following published papers---(examples)
1. Bioactive constituents from the leaves of Melastoma malabathricum L--- by Deny sushanti
2. In vitro antioxidant potential and phytochemical profiling of Melastoma malabathricum leaf water extract—by Oke
3. Ethnobotanical, Phytochemical, and Pharmacological Aspects of Melastoma sp… by samad
4. Review of Pharmacognostic Features, Phytochemical Constituents and Pharmacological Actions of Melastoma malabathricum LINN (Melastomaceae)==by Danladi
